# Prevalence of Diabetic Retinopathy in Type 1 and Type 2 Diabetes Mellitus Patients in North-East Poland

**DOI:** 10.3390/medicina56040164

**Published:** 2020-04-06

**Authors:** Wojciech Matuszewski, Angelika Baranowska-Jurkun, Magdalena M. Stefanowicz-Rutkowska, Robert Modzelewski, Janusz Pieczyński, Elżbieta Bandurska-Stankiewicz

**Affiliations:** 1Clinic of Endocrinology, Diabetology and Internal Medicine, Department of Internal Medicine, School of Medicine, Collegium Medicum, University of Warmia and Mazury, 10-561 Olsztyn, Poland; angelika_b1990@o2.pl (A.B.-J.); m.m.stefanowicz@gmail.com (M.M.S.-R.); robur23@interia.eu (R.M.); bandurska.endo@gmail.com (E.B.-S.); 2Ophthalmology Clinic, School of Medicine, Collegium Medicum, University of Warmia and Mazury, 10-561 Olsztyn, Poland; janusz.pieczynski@op.pl

**Keywords:** diabetes mellitus, microangiopathy, diabetic retinopathy, prevalence

## Abstract

*Background and Objectives*: The global epidemic of diabetes, especially type 2 (DM2), is related to lifestyle changes, obesity, and the process of population aging. Diabetic retinopathy (DR) is the most serious complication of the eye caused by diabetes. The aim of this research was to assess the prevalence of diabetic retinopathy in type 1 and type 2 diabetes mellitus patients in north-east Poland. *Materials and Methods*: The eye fundus was assessed on the basis of two-field 50 degrees color fundus photographs that showed the optic nerve and macula in the center after the pupil was dilated with 1% tropicamide. *Results*: The experimental group included 315 (26%) patients with type 1 diabetes mellitus (DM1) and 894 (74%) patients with DM2. DM1 patients were diagnosed with DR in 32.58% of cases, with non-proliferative diabetic retinopathy (NPDR) in 24.44% of cases, proliferative diabetic retinopathy (PDR) in 1.59% of cases, diabetic macular edema (DME) in 5.40% of cases, and PDR with DME in 0.95% of cases. DR was found in DM2 patients in 23.04% of cases, NPDR in 17.11% of cases, PDR in 1.01% of cases, DME in 4.81% of cases, and PDR with DME in 0.11% of cases. *Conclusions:* The presented study is the first Polish study on the prevalence of diabetic retinopathy presenting a large group of patients, and its results could be extrapolated to the whole country. Diabetic retinopathy was found in 25.48% of patients in the whole experimental group. The above results place Poland within the European average, indicating the quality of diabetic care offered in Poland, based on the number of observed complications.

## 1. Introduction

Diabetes is the only non-communicable disease considered by the World Health Organization (WHO) to be an epidemic. At present, there are 463 million diabetic patients in the world, and this number is expected to exceed half a billion before 2045, with current forecasts suggesting that the global number of diabetic patients will reach 700 million. Taking into account other epidemiological data, the number of diabetic patients has doubled every 20 years since 1945. Currently, in Europe there are 59 million diabetic patients, and in Poland there are 2.34 million adult diabetic patients [1,2]. Type 2 diabetes mellitus (DM2) amounts for 85%–95% of diabetes cases, and a significant increase in the number of cases has been recorded in developing countries, including Poland [3,4]. Type 1 diabetes mellitus (DM1) constitutes 15%–20% of cases, and it is found particularly often in European populations. In Poland, the incidence rate of DM1 is 10.2/100,000 people/year, with an increasing frequency, placing Poland among countries with an average DM1 incidence [5,6]. Diabetes as an epidemic of the modern world leads to long-standing consequences related to human suffering and economic costs. The disease is accompanied by disability and a high death rate, which is caused by chronic vascular complications. Diabetic retinopathy (DR) affects over one-third of all people with diabetes. DR is the most severe complication in the eye caused by diabetes; it may occur along with the diagnosis of the disease and constitutes 80% of causes of vision loss in this population [1,7]. So far, no data concerning the prevalence of diabetic retinopathy in Poland has been available, as the country has not had any national screening program to determine the number of people with this condition. For years, such programs have functioned in a number of countries, providing significant information and depicting the range of medical, social, and economic problems related to diabetic eye disease.

## 2. Materials and Methods

### 2.1. Sample Population

In the years 2012–2016, screening for diabetic retinopathy was conducted in the Warmian-Masurian region, the fourth largest region in Poland, with 3.8% of the population of the country, which has not changed since 2010. According to the National Health Fund, the prevalence of diabetes in the Warmian-Masurian Voivodeship is 3.5% [8]. The experimental group consisted of adult DM1 and DM2 patients who were diagnosed with diabetes according to the WHO criteria [9]. The study encompassed patients treated in the Endocrinology, Diabetology and Internal Diseases Clinic of the University of Warmia and Mazury in Olsztyn, Voivodeship Diabetological Centre, local diabetological centers, and by randomly chosen family physicians running their practices in the Warmian-Masurian Voivodeship. The study was conducted following the approval number 10/2010, date 25.03.2010 of the Bioethics Committee of the Faculty of Medicine University of Warmia and Mazury in Olsztyn, Poland.

### 2.2. Methods of Assessing the Fundus of the Eye

The research was carried out by a technician trained at the Retinal Screening Center at Heartlands Hospital in Birmingham, UK. Eye fundus photographs were evaluated by an experienced ophthalmologist specialist (at place) certified in the framework of the DIRECT Study (Diabetic Retinopathy Candesartan Trials) and at the Retinal Screening Center at Heartlands Hospital in Birmingham, UK.

Only the fundus of the eye was assessed on the basis of two-field 50 degrees color fundus photographs that showed the optic nerve and macula in the center after the pupil had been dilated with 1% tropicamide. The photographs were taken with a Topcon TRC NW8 Fundus Camera (Topcon Medical Systems, Inc., Oakland, NJ, USA). Severity of diabetic retinopathy was assessed according to the criteria of the International Clinical Classification for Diabetic Retinopathy, and the following stages were enumerated: no diabetic retinopathy; non-proliferative diabetic retinopathy (NPDR); proliferative diabetic retinopathy (PDR); diabetic macular edema (DME) [10,11]. No DR was considered as no observed abnormalities. NPDR was considered as one or more of the following: microaneurysms, intraretinal hemorrhages, venous beading, and/or prominent intraretinal microvascular abnormalities and no signs of proliferative retinopathy. PDR was considered as one or more of the following: neovascularization and/or vitreous/preretinal hemorrhage.

### 2.3. Statistical Analysis

Statistical analysis was performed with the use of STATISTICA 10 PL statistics package (StatSoft Polska, Kraków, Poland). ANOVA (*F*-test and post hoc tests) or the Student’s *t*-tests were applied in order to test the proposed hypotheses comparing mean values in the analyzed groups. If the required assumptions were not met, U-Mann–Whitney non-parametric tests were applied for two groups, or the Kruskal–Wallis test was used if there were more than two groups. The significance level *p* = 0.05 was determined for all the tests.

## 3. Results

### 3.1. Description of the Experimental Group

The study was carried out in the years 2012–2016, and it encompassed 1209 patients, 53% of whom were women. There were 315 (26%) DM1 patients, who were on average 37.0 (13.55) years old with 12.3 (9.28) years of diabetes, and 894 (74%) DM2 patients, who were on average 61.2 (11.13) years old with 10.5 (8.09) years of diabetes. Of the 1209 patients, 44% of the patients lived in cities, 29% lived in towns, and 27% lived in the countryside. DM1 patients were treated with insulin only, while 39% of DM2 patients were given oral antidiabetic medications, 23% were given insulin, and 38% were given a combined treatment. 

### 3.2. Prevalence of Diabetic Retinopathy in the Experimental Group

Diabetic retinopathy was found in 25.48% of patients, with NPDR (non-proliferative diabetic retinopathy) in 19.02%, PDR (proliferative diabetic retinopathy) in 1.16%, DME (diabetic macular edema in 4.96%, and PDR (proliferative diabetic retinopathy) with DME (diabetic macular edema) in 0.33% of the group. In DM1 patients, DR was diagnosed in 32.58% of cases, including NPDR (non-proliferative diabetic retinopathy) in 24.44%, PDR (proliferative diabetic retinopathy) in 1.59%, DME (diabetic macular edema) in 5.40%, and PDR (proliferative diabetic retinopathy) with DME (diabetic macular edema) in 0.95% of patients. In DM2 patients, DR was found in 23.04% of cases, including NPDR (non-proliferative diabetic retinopathy) in 17.11%, PDR (proliferative diabetic retinopathy) in 1.01%, DME (diabetic macular edema) in 4.81%, and PDR (proliferative diabetic retinopathy) with DME (diabetic macular edema) in 0.11% of cases. NPDR (non-proliferative diabetic retinopathy) and DME (diabetic macular edema) were found significantly more often in DM1 patients than in DM2 patients (*p* < 0.01) (Figure 1, Table 1)

Diabetic retinopathy was diagnosed significantly more frequently in men than in women: 158 (27.72%) vs. 150 (23.47%) (*p* = 0.011). However, no statistically significant differences were found between sexes in terms of the prevalence of the different stages of retinopathy (Table 2).

## 4. Discussion

The presented research results concern the population of the Warmian-Masurian Voivodeship. However, because of the number of patients and their random selection, the results can be extrapolated to the whole country. Considering the whole experimental group, diabetic retinopathy was found in 25.48% of patients, with NPDR in 19.02%, PDR in 1.16%, DME in 4.96%, and PDR with DME in 0.33%. Among DM1 patients, DR was found in 32.58% of cases, including NPDR in 24.44%, PDR in 1.59%, DME in 5.40%, and PDR with DME in 0.95%. In DM2 patients, DR was diagnosed in 23.04% of cases, including NPDR in 17.11%, PDR in 1.01%, DME in 4.81%, and PDR with DME in 0.11%. In Russia, the prevalence of DR was 45.9%; in more recent studies concerning DM1 patients, it was 35%; and in DM2 patients, it was 16.67% [12,13]. In Germany, in the years 2002–2004, a population of 5596 diabetic patients was analyzed and DR was found in 10.6% of the patients, with NPDR in 10%, PDR in 0.5%, and DME in 0.85% [14]. On the basis of the most recent German data from 2015, it can be concluded that the prevalence of diabetic retinopathy in the group of 15,000 diabetic patients amounted to 21.7% and that the prevalence of DME accounted to 2.3% [15]. In France, the prevalence of DR in DM2 patients was determined to be 33%, including PDR in 3.3% and DME in 5.6% [16,17]. However, in a subsequent screening study DR-PHDIAT (Ophthalmology Diabetes Telemedicine) conducted in France on a population of 13,777 diabetic patients, the prevalence of DR was determined to be 23.4%, with DME in 3.4% and PDR in 0.5%. These results are very close (especially the prevalence of DR and DME) to those presented in this study [18]. In the classical United Kingdom Prospective Diabetes Study (UKPDS) conducted in 23 centers in England on a population of more than five thousand patients diagnosed with DM2, retinopathy was found in 35%–39% of cases [19]. Another study published in the UK in 2000 encompassed ten thousand diabetic patients and showed the prevalence of retinopathy at the level of 16.5% [20]. In another British study, the prevalence of retinopathy among patients with type 1 diabetes was 45.7%, (PDR: 3.7%; sight-threatening DR (STED): 16.4%), and in the DM2 group it was 25.3% (PDR: 0.5%; STED: 6.0%) [21]. In Wales in 2005–2008, a large population study was carried out on a group of 91,393 patients, and diabetic retinopathy was diagnosed in 32.4% of patients, including non-sight-threatening retinopathy in 29.0% and sight-threatening retinopathy in 3.4%. In Scottish studies, the prevalence of retinopathy was determined to be 28.1%. In Wales, similar to the Polish research results presented in this article, the prevalence of diabetic retinopathy was higher in DM1 patients, amounting to 56.3% of the patients, with non-sight-threatening retinopathy in 45.1%, and sight-threatening retinopathy in 11.2%, while DME amounted to 4.2%. Among DM2 patients, retinopathy was diagnosed in 30.9% of patients, including non-sight-threatening in 28.1%, sight-threatening in 2.9%, and maculopathy in 1.4% [22]. Data from Sweden published in 1998 indicated a 28% prevalence of diabetic retinopathy, and in subsequent more precise screening research DR was found in 41.8% of DM1 patients, with sight-threatening retinopathy in 12.1% of cases. In DM2 patients, the results were, respectively, 27.9% and 5.0% [23,24]. These studies confirm more frequent eye complications in the DM1 group of patients, which is also the case in the Polish studies presented in this paper. In Norway (in 2007–2008), the prevalence of DR was similar to that in the presented study, and amounted to 26.8%, with a slightly lower percentage of maculopathy, which was 3.9%, whereas in the Netherlands, DR was found in 35% of DM2 patients [25,26]. In a study comprising DM2 patients from Italy, a considerably lower prevalence of retinopathy was observed as compared with the current study, namely 9.8%, including NPDR in 4.2%, PDR in 4.2%, and DME in 1.3% [27]. In the years 2009–2014, in the first screening program for diabetic retinopathy in Portugal, similar to the situation in Italy, DM2 patients were found to suffer from diabetic retinopathy less frequently than in the current study. DR was diagnosed in 8584 patients (16.3%), with NPDR in 14.5% of cases, PDR in 1.8%, and maculopathy in 1.4% [28]. In Spain, despite a similar geographical location, the results were decidedly different and similar to those presented in the current study. This means that diabetic retinopathy in DM1 patients was 36.47% and DME was 5.73%, while in DM2 patients this was, respectively, 26.11% and 6.44% [29]. While summarizing the data from numerous European screening programs, another study needs to be mentioned: the Prospective Complications Study (EURODIAB)—a European multicenter trial encompassing DM1 patients, which was the first trial to determine the prevalence of DR in DM1 patients at the level of 35.9%, with PDR in 10.8% [30,31]. The EURODIAB study presented considerable disparities regarding DR prevalence in different countries. The prevalence of DR in DM1 patients determined in the current study is 32.58%, which places Poland within the European average. In the meta-analysis of 35 studies conducted all over the world in the years 1980–2008, altogether, including 22,896 people with diabetes, the prevalence of DR was determined to be 34.6%, out of which 6.96% was PDR, 6.81% was DME, and 10.2% was sight-threatening diabetic retinopathy (STDR). Similar to the current study, the prevalence of retinopathy among DM1 patients was higher than among DM2 patients. Similar to other countries, no differences in the prevalence of DR, PDR, and DME were found between sexes [32]. In the Wisconsin Epidemiologic Study of Diabetic Retinopathy (WESDR) in the USA, DR was found in 50.3% of patients, which is a significantly larger percentage than in the presented experimental group. However, just like in the current study, retinopathy occurred more often in the group that underwent insulin treatment. The latter was administered in 53% of patients with eye complications, while 13% of patients received antidiabetic medications, and 34% received a combined treatment. In WESDR, insulin was used in 62% of patients, while antidiabetic drugs were given to 36% of patients [33,34]. A high prevalence of diabetic retinopathy, that is 54.2%, was determined in the Diabetes Control and Complication Study (DCCT), which comprised DM1 patients from the USA and Canada [35,36]. Other data was published in 2010 on the basis of a study conducted in the USA in the years 2005–2008 in a group of 1006 diabetic patients. The prevalence of DR was determined at the level of 28.5%, and it was (statistically significantly) more often found in men than in women (31.6% and 25.7%, respectively, which was statistically significant), just like in the current study, where the prevalence was 27.72% and 23.47%, respectively [37]. In the current study, no differences between sexes were determined in regard to the occurrence of various stages of DR. The Canadian study of 2010 revealed a prevalence of 27.2% for DR, with 24.95% for NPDR, 2.3% for PDR, and 2.0% for DME [38]. Screening in Australia did not differ from that in Canada and the USA, with results of retinopathy ranging from 23.4% to 29.3%, which are comparable with those found in this study [39,40,41]. Data concerning the prevalence of DR in poorly developed countries are disparate. For instance, in Bangladesh DR appears in 19% of cases, whereas in India it appears in 17%–22% of cases, with 30.3% in Cambodia, 37% in Iran, and 63% in South Africa [42,43,44,45,46,47] (Table 3).

Differences in the prevalence of DR have been observed to also depend on the degree of urbanization. In the studies conducted in China, a much higher percentage of retinopathy and maculopathy was observed in urban areas (43% and 3.5%, respectively) as compared to rural areas (73% and 2.6%, respectively) [48]. Other Chinese researchers pointed out even greater disproportions dependent on the place of living: 17.6% of DR cases among city dwellers and 43.1% among the countryside population [49]. In the presented study, 37% of patients with retinopathy lived in cities, 35% lived in towns, while 28% lived in the countryside. The presented review of literature indicates that the data on the prevalence of diabetic retinopathy is not consistent. There are numerous factors that may have an effect on diabetic retinopathy screening results. The most important of them is the degree and possibility of diagnosing diabetes. In highly developed countries, such as the UK, it is estimated that 25% of the cases of diabetes are undiagnosed, while in the USA, it is estimated that 23.8% of the cases are undiagnosed [50,51]. In comparison, in poor developing countries the number of undiagnosed cases are estimated to be much higher, reaching 52% in India, 66% in Cambodia, 70% in Ghana, and 80% in Tanzania [52,53,54]. There are also other important factors, such as social situation, access to medical care, the experience of medical staff, and many others. Yet still it seems that economic factors play the greatest role.

## 5. Conclusions

Diabetes constitutes a great challenge not only medically but also socially and economically. The greatest problem results from vascular complications, which not only lower the quality of life of diabetic patients but also generate huge social costs. The presented study is the first Polish study on the prevalence of diabetic retinopathy presenting a large group of patients, and its results can be extrapolated to the whole country. Diabetic retinopathy in the whole experimental group amounted to 25.48%. This result places Poland within the European average, which is indicative of the quality of diabetes care offered in Poland, as it is assessed on the basis of the number of complications. One can only hope that this screening for diabetic retinopathy can become a pilot program for other regions of Poland and that, just like in other countries, it will contribute to the establishment of the national screening program for diabetic retinopathy.

## Figures and Tables

**Figure 1 medicina-56-00164-f001:**
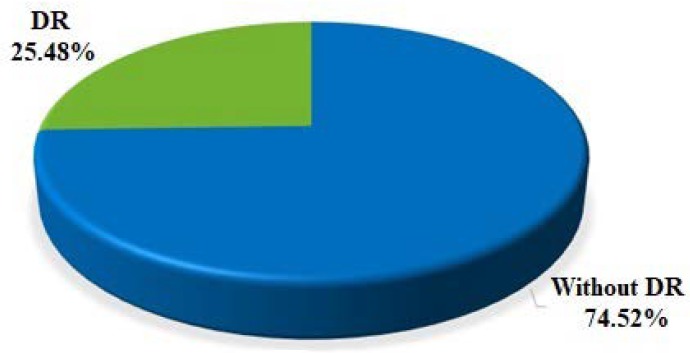
Prevalence of diabetic retinopathy (DR) in the experimental group.

**Table 1 medicina-56-00164-t001:** Stages of advancement of diabetic retinopathy in the experimental group with reference to type of diabetes.

	Experimental Group *n* (%)			*p*
	Total	DM1	DM2	
DR	308 (25.48)	102 (32.58)	206 (23.04)	0.001
• NPDR	230 (19.02)	77 (24.44)	153 (17.11)	0.001
• PDR	14 (1.16)	5 (1.59)	9 (1.01)	ns
• DME	60 (4.96)	17 (5.40)	43 (4.81)	0.002
• PDR and DME	4 (0.33)	3 (0.95)	1 (0.11)	ns
Without DR	901 (74.52)	213 (67.62)	688 (76.96)	0.001

DM1: type 1 diabetes mellitus; DM2: type 2 diabetes mellitus; DR: diabetic retinopathy; NPDR: non-proliferative diabetic retinopathy; PDR: proliferative diabetic retinopathy; DME: diabetic macular edema, ns: not significant.

**Table 2 medicina-56-00164-t002:** Prevalence of diabetic retinopathy depending on the advancement stage with reference to sex.

	Experimental Group *n* (%)			*p*
	Total	Women	Men	
DR	308 (25.48)	150 (23.47)	158 (27.72)	0.011
NPDR	230 (19.02)	117 (18.31)	113 (19.82)	ns
PDR	14 (1.16)	7 (1.10)	7 (1.23)	ns
MD	60 (4.96)	23 (3.60)	37 (6.49)	ns
PDR and DME	4 (0.33)	3 (0.47)	1 (0.18)	ns
Without DR	901 (74.52)	489 (76.53)	412 (72.28)	0.011

**Table 3 medicina-56-00164-t003:** Prevalence of diabetic retinopathy in different countries.

Country	Prevalence of Diabetic Retinopathy %			References
	DM1	DM2	Total	
Germany	n/a	n/a	21.7	[15]
Russia	35	16.67	45.9	[12,13]
France	n/a	n/a	23.4	[18]
UK	n/a	35–39	n/a	[19]
UK	45.7	25.3	n/a	[21]
Wales	56.3	45.1	n/a	[22]
Sweden	n/a	n/a	28	[23]
Sweden	41.8	27.9	n/a	[24]
Norway	n/a	n/a	26.8	[25]
Netherlands	n/a	35	n/a	[26]
Italy	n/a	9.8	n/a	[27]
Portugal	n/a	16.3	n/a	[28]
Spain	36.47	26.11	n/a	[29]
USA	n/a	n/a	50.3	[33]
USA, Canada	54.2	n/a	n/a	[36]
Canada	n/a	n/a	27.2	[38]
Australia	n/a	n/a	23.4–29	[39,40,41]
Bangladesh	n/a	n/a	19	[42]
India	n/a	n/a	17–22	[43,44]
Cambodia	n/a	n/a	30.3	[45]
Iran	n/a	n/a	37	[46]
South Africa	n/a	n/a	63	[47]

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
