# Peer review of "Prevalence of Diabetic Retinopathy in Type 1 and Type 2 Diabetes Mellitus Patients in North-East Poland"

_medicina, 2020, doi:10.3390/medicina56040164_

Round 1
Reviewer 1 Report
Peer review for the manuscript medicina-728734
Title: "Prevalence of Diabetic Retinopathy in Type 1 and Type 2 Diabetes Mellitus Patients in North-East Poland"
Minor comments for the authors:
I suggest that acronyms such as NPDR, PDR reported in the result section should be spelled out.
In the discussion section many studies are reported describing different percentages of DR in different populations. Maybe I suggest to describe the main numerical findings (i.e percentages) of the different studies in a table.
Author Response
Dear Reviewer,
Thank you for your time and all comments. In the results I explained the acronyms in the result section and I added a table with the prevalence of diabetic retinopathy in different countries.
Yours faithfully
Wojciech Matuszewski

Reviewer 2 Report
The text neds extensive English language editing - some errors are indicated in the attached pdf file. The methods are not cleraly described-who perfomed the screening-ophthalmologist or technician, who performed analysis-was it reading center or was it done at place? Was OCT also performed or only fundus photograph was done? Could you describe criteria for diabetic retinopathy and macular edema more extensively? Could you provide any pictures? Was visual acuity measured or not?

Author Response
Dear Reviewer,
Thank you for your time and all comments.
1.Who perfomed the screening-ophthalmologist or technician? -The research was carried out by a technician (trained diabetologic nurse- technical training performed at the Retinal Screening Center, Heartlands Hospital in Birmingham UK). Eye fundus photographs were evaluated by an experienced ophthalmologist specialist (at place) certified in the framework of the DIRECT Study (DIabetic REtinopathy Candesartan Trials) and at the Retinal Screening Center at Heartlands Hospital in Birmingham UK.
2. Who performed analysis-was it reading center or was it done at place? - At place.
3. Was OCT also performed or only fundus photograph was done? - only fundus photograph.
4. Could you describe criteria for diabetic retinopathy and macular edema more extensively?- status of diabetic retinopathy was assessed according to the criteria of the International Clinical Classification for Diabetic Retinopathy- by the American Academy of Ophthalmology.
5. Could you provide any pictures? - I can attach an example of a fundus photo.
6. Was visual acuity measured or not?- No, only fundus photograph.
Language correction in progress.
Yours faithfully
Wojciech Matuszewski

Round 2
Reviewer 2 Report
No comments